# Research on the Equity and Optimal Allocation of Basic Medical Services in Guangzhou in the Context of COVID-19

**DOI:** 10.3390/ijerph192214656

**Published:** 2022-11-08

**Authors:** Jin Li, Jie Li, Jian Huang

**Affiliations:** School of Geography and Remote Sensing, Guangzhou University, Guangzhou 510006, China

**Keywords:** health care, basic medical services, accessibility, equity, resource allocation, G2SFCA

## Abstract

Optimizing the allocation of basic medical services and ensuring their equity are necessary to improve the ability to respond to public health emergencies and promote health equity in the context of COVID-19. This study aims to analyze the equity of Guangzhou’s basic medical service and identify areas where health resources are relatively scarce. The spatial distribution and patterns of basic medical services were analyzed using kernel density analysis and standard deviation ellipse. The equity was analyzed using the Gini coefficient and Lorenz curve in terms of population and geographical area, respectively. Considering the medical demand and supply sides, the Gaussian two-step floating catchment area method was used to analyze the accessibility to different levels of medical institutions. The kernel density analysis and standard deviation ellipse showed that the spatial distribution of medical and health resources in Guangzhou is unevenly distributed, and high-level hospitals and medical resources are mainly concentrated in the centrum. From the perspective of population, Guangzhou’s medical equity is generally reasonable. The accessibility of medical institutions differs with different levels, and the tertiary medical institutions have the best accessibility, while the unclassified, primary, and secondary medical institutions generally have lower accessibility. The accessibility of districts in Guangzhou varies greatly. Areas in the center are most accessible to basic medical services, while accessibility in outskirt areas has gradually decreased. Conclusion: The quantity of per capita medical and health resources in Guangzhou, as evidenced by basic medical services, is sufficient, but the spatial distribution is unequal. The developed city center enjoys more adequate healthcare resources than the distant suburbs. Primary healthcare should be built, especially in distant suburbs, to strengthen basic medical service equity in Guangzhou.

## 1. Introduction

Ensuring everyone has equal opportunity and can access basic medical services (BMS) whenever one needs is included in goal three of the United Nations 2030 global sustainable development goals, good health, and well-being [1]. To achieve this goal, the BMS’s role cannot be over-emphasized [2], as BMS provides effective, safe, and necessary medical treatment at an affordable price, providing the most rudimentary health care services for all.

With the acceleration of urban construction in China has been rapid, and many rural migrant workers rush to big cities. However, the construction of BMS lags far behind urban construction. As a result, the problems of insufficient medical and health resources, uneven distribution, and regional inequality have become more prominent [3]. In addition, since the outbreak of COVID-19 at the end of 2019, COVID-19 has seriously affected people’s health in 31 provinces, autonomous regions, and cities in China, as well as the social stability of families [4]. Furthermore, it has further strained medical resources with unprecedented medical needs far exceeding the service capacity of existing hospitals [5]. Besides, the large size and frequent mobility of China’s population have provided conditions for the insidious spread of COVID-19, forcing the country to be constantly in a state of alert for epidemic prevention and control.

The tiered medical care system is an essential part of BMS. It effectively reduces medical costs and alleviates the problem of “difficult and expensive access to medical care” in China [6]. Corresponding medical treatment can be carried out according to the severity of the patient’s illness. Therefore, the tiered medical care system maximizes the response to many confirmed COVID-19 patients and asymptomatic infected patients. Besides, it plays a vital role in guaranteeing routine medical services, ensuring treatment for different diseases, strengthening regional resource integration, avoiding resource squeeze, and maximizing resource benefits [7]. However, the current situation of medical resources supply in China is in an “inverted triangle” pattern. High-quality medical resources are mainly concentrated in large cities and high-level hospitals, with fewer high-quality medical practitioners and limited service capacity in primary care institutions. Most medical resources and patients are concentrated in tertiary hospitals [8]. At the current stage, regional development and the imbalance of medical and health conditions are still the main problems restricting the equity of medical services in China. In this context, how to reasonably allocate and optimize the limited health resources is an urgent problem to be solved to enhance the equity of BMS and improve people’s health.

Medical equity means that there is no inequity in access to health for each individual. Ensuring the equity and accessibility of BMS is an important aspect of achieving the goal of basic medical security, a vital component of reflecting social equity and building a harmonious socialist society. In addition to China, the equity of medical resource allocation in other countries has also received the same attention. Scholars from the United States [9,10], the United Kingdom [11], Turkey [12], Iran [13], Kenya [14], and other countries have investigated and analyzed the allocation of health care resources in their respective countries. They also proposed recommendations and methods to optimize the allocation of health care resources. Existing studies found that the inequitable distribution of healthcare resources is typical in these countries, and the factors that affect equity and accessibility are gender [15], education level [15], economic status [16], ethnicity [17], the nature of hospital operations [18], consultation time [19], etc. Furthermore, existing studies have revealed that the distribution of medical resources has significant regional characteristics [20], especially in rural [11] and remote areas [21]. The allocation of medical and health care resources is seriously inadequate in areas with low economic development, such as rural and remote areas.

Chinese scholars have mainly studied the spatial and temporal characteristics of health care resource allocation based on urban–rural and regional differences. The allocation of medical and health resources in China’s urban and rural areas has a reasonable per capita distribution, but there is polarization. There is apparent uneven geographical distribution, a lack of medical and health resources in less developed regions, and insufficient health investment [22,23,24]. In addition, many high-quality medical resources are concentrated in large cities and central areas [25,26,27]. In previous studies, the main methods of measuring equity are the Gini coefficient [22,28], the Lorenz curve [29], and the aggregation index [2]. The main methods for measuring spatial accessibility are the nearest distance method [18], the gravity model method [30], the cumulative opportunity method [31], and the two-step floating catchment area method (2SFCA) [32,33]. Among the above methods, the 2SFCA method is widely used to calculate the accessibility of health care services [2,11,33,34], because it integrates the influence of both the supply side and the demand side, which can calculate the accessibility more comprehensively. However, traditional 2SFCA ignores the part of the effect that as the distance increases, the attractiveness of health care institutions to residents decreases, i.e., no attenuation within the service threshold [35]. Many scholars have improved 2SFCA to address this shortcoming according to their research’s problems, objects, and areas. Weight assignment and superposition of the service scope of hierarchical medical facilities were used to evaluate the medical accessibility of mountainous cities [36]. An optimized version of 2SO4SAI (two-step optimization for spatial accessibility improvement) was proposed to investigate the accessibility of medical care in rural areas [37]. A combination of the closest proximity method and a modified version of the 2SFCA method were used to analyze the differences in medical accessibility between minority and non-minority areas [17]. Gaussian function was introduced to model the effects of distance decay to analyze the accessibility of medical institutions above the second level [26]. In summary, existing studies have mainly measured the accessibility of BMS and analyzed the factors affecting equity, but less from the perspective of graded medical institutions to consider the equity differences of different levels of medical institutions.

This study analyzed the spatial distribution characteristics and accessibility of different levels of medical institutions in the basic medical service system. In addition, a correlation study was conducted to effectively distinguish the areas where medical resources are scarce and provide corresponding guidance and suggestions for resource optimization.

The study area of this paper is Guangzhou, a first-tier coastal city in southern China. As a mega-city, Guangzhou cannot meet its sizeable resident population’s demand for medical care despite its abundant medical resources. Combined with the heavy pressure of epidemic prevention and control, this has created new pressure on the city’s medical and health services. Therefore, the objectives of this paper are to identify the current situation of BMS in Guangzhou and to analyze the accessibility of BMS in the context of regular epidemic prevention and control.

## 2. Materials and Methods

### 2.1. Study Area

Guangzhou is located in the south-central part of Guangdong Province, with geographical coordinates of 112°57′ E–114°3′ E, and 22°26′ N–23°56′ N, the northern edge of the Pearl River Delta. As of 2021, the city has 11 districts: Liwan District (LW), Yuexiu District(Y X), Haizhu District (HZ), Tianhe District (TH), Baiyun District (BY), Huangpu District (HP), Panyu District (PY), Huadu District (HD), Nansha District (NS), Zengcheng District (ZC), and Conghua District (CH), of which YX, LW, and HZ are the old urban areas, and TH is the current center of Guangzhou. Guangzhou is one of the first national historical and cultural cities and the birthplace of Guangfu culture. It is the capital of Guangdong Province, a sub-provincial city, a national central city, a mega-city, the core city of the Guangzhou metropolitan area, an important central city, and an international trade center. In addition, Guangzhou is the headquarters of China’s Southern War Zone Command, the southern gate of China to the world, the central city of the Guangdong-Hong Kong-Macao Bay Area and the Pan-Pearl River Delta Economic Zone, and a hub city of the Belt and Road.

In recent years, Guangzhou has continuously improved the quality of its medical infrastructure. Guangzhou has enhanced the professionalism of its medical practitioners, optimized the allocation of medical resources, and strived to meet its residents’ medical and health needs in the past few years. According to the Guangdong Provincial Statistical Yearbook, during 2016–2020, the number of medical institutions in Guangzhou increased from 3806 to 5550, the number of beds increased from 87,959 to 101,640, the number of health workers increased from 16,6537 to 21,612, and the number of licensed physicians increased from 46,791 to 62,329.

### 2.2. Data Source

The following research data are used to analyze Guangzhou’s spatial distribution characteristics and accessibility of medical and health services: ① Hospital vector data. To analyze the current situation of BMS in Guangzhou in a more targeted manner, the medical institutions referred to later in this study are limited to the type of hospitals. Relevant data are obtained from the Guangzhou Municipal Health Commission and Guangzhou City Life Map app. Their attributes include the address, level, and the number of beds in each hospital. The geocoding function of Baidu Map API was invoked to assign and convert the coordinates of each medical facility. A total of 322 medical facilities were obtained in the study area, with hospital types covering general hospitals, specialty hospitals, and plastic and cosmetic hospitals. The hospital levels cover primary, secondary, tertiary, and unclassified, including 62 primary hospitals, 98 secondary hospitals, 74 tertiary hospitals, and 88 unclassified hospitals. The distribution of medical facilities is shown in Figure 1. ② Population density data in 2020 from the WorldPop dataset (https://www.worldpop.org/, accessed on 9 September 2022) with a resolution of 1000 m.

### 2.3. Research Methodology

#### 2.3.1. Kernel Density and Standard Deviation Ellipse Analysis

The spatial distribution characteristics of the medical institutions studied in this paper are analyzed mainly by kernel density analysis and standard deviation ellipse analysis. Kernel density analysis is a research method used to discover the clustering and discrete distribution characteristics of point data, which can effectively reflect the geographic decay effect and is often used to explore the distribution pattern of data [38]. It can transform the discrete point data into a continuous density map. With the element point as the center and the selected threshold radius, the closer to the center of the element point, the greater the weight given to the density grid due to spatial autocorrelation. The calculation formula is expressed as:(1)$$f\left(x\right)=\sum_{i=1}^{n}\frac{1}{h^{2}}k\left(\frac{a_{x}-a_{i}}{h}\right)$$
where *f*(*x*) is the kernel density at the *x*th point, *h* is the decay distance threshold, *n* is the total number of samples, *k* is the kernel function, and $a_{x}-a_{i}$ is the distance between sample point *i* and the center point *x*. The average impact range of the combined hospitals was set to a distance threshold of 5000 m for the analysis.

The standard deviation ellipse (SDE) is very sensitive to the distribution and direction of the data. Therefore, it can effectively express the distribution direction and extent of the study object. SDE quantitatively describes the spatial distribution of geographic elements through parameters such as center, long, and short axes, directional angle, and flatness [39]. The center of the SDE characterizes the spatial distribution of geographic elements. The center of the SDE shows the center of gravity of the study object and reflects the spatial distribution of the study object in the geographic center of the study area. The longer the long axis, the more obvious the direction of the object of study is, and the shorter the short axis, the more concentrated the distribution of the object of study is. The directional angle is the clockwise rotation angle with the long axis prevailing at 0° due north, which is used to show the distribution direction of the research object. The flat rate is the ratio of the long and short axes, which is used to reflect the concentration of the distribution of the research object. The larger the flat rate, the more concentrated the distribution of the research object, and vice versa, the more dispersed.

#### 2.3.2. Lorenz Curve and Gini Coefficient

Lorenz curve and Gini coefficient were used to evaluate and analyze the equity of resource allocation of medical institutions in Guangzhou. The Lorenz curve was proposed by the American statistician Lorenz in 1907 and was initially commonly used to explore the equity of income distribution [29]. The Lorenz curve is based on the connotation of social equity, and income distribution is similar to the distribution of public resources [34]. In recent years, the Lorenz curve has been widely used in the social resource allocation equity field, using the size of the area between the curve and the absolute equity line to reflect the equity of the allocation of the study object. The larger the area means the worse the equity, and vice versa, the better. In the Lorenz curve graph, the diagonal line is the absolute equity line, and the closer to the absolute equity line indicates the better equity.

The Gini coefficient was proposed by Italian statistician Gini in 1922 [10]. It is calculated based on the Lorenz curve, and the value range is 0~1. The larger the value, the worse the equity of the research object configuration, and vice versa, the better it is, and 0.4 is usually taken as the “warning line” of equity internationally [10,27,29]. The value is less than 0.2 is fair, between 0.2 and 0.3 is relatively fair, between 0.3 and 0.4 is relatively fair, 0.4 to 0.5 is unfair, 0.5 to 0.6 is a huge gap, and more than 0.6 is a highly unfair state [21]. Based on existing research [23], the Gini coefficient chosen in this paper is calculated as follows:(2)$$G=1-\sum_{i=1}^{n}(X_{i}-X_{i-1})\left(Y_{i}+Y_{i-1}\right)$$
where *G* is the size of the Gini coefficient, *n* is the number of surveyed districts, *i* is the ranking number of resource possession per capita, $X_{i}$ is the cumulative percentage of the population or geographical area of each district, and $Y_{i}$ is the cumulative percentage of resources of medical institutions, where $X_{0}$ = 0 and $Y_{0}$ = 0.

#### 2.3.3. G2SFCA

The 2SFCA considers the demand and supply sides, respectively, and can analyze the accessibility of medical institutions more comprehensively [2,33,37]. The 2SFCA with the Gaussian function as distance decay function is one of the most commonly improved 2SFCA [26,30,40]. Based on the previous studies [40], the accessibility of medical institutions in Guangzhou is analyzed in the following steps.

In the first step, for the supply point *j* of the medical institution, a search domain *J* is established with the radius of the limit distance *d*_0_ of the road network for people to go to the hospital. All the populations in the search domain *j* are aggregated, and weights are assigned according to the distance decay law using Gaussian function. Next, these weighted populations were summed and aggregated to calculate the supply–demand ratio *R_j_*. The ratio of:(3)$$R_{j}=\frac{S_{j}}{\sum_{k\in \left\{d_{kj}\leq d_{0}\right\}}G\left(d_{ij}\right)D_{k}}$$
where $D_{k}$ is the population size of each demand cell *k,* and $d_{kj}$ is the distance between locations *k* and *j*. Cell *k* needs to fall within the search domain (i.e., $d_{kj}$ ≤ $d_{0}$). $S_{j}$ is the number of beds in medical facility *j*. $G\left(d_{ij}\right)$ is the Gaussian function considering the decrease of accessibility with increasing distance, which can be expressed in the specific form of:(4)$$G\left(d_{ij}\right)=\frac{e^{-\frac{1}{2}*\left(\frac{d_{ij}}{d_{0}}\right)}-e^{-\frac{1}{2}}}{1-e^{-\frac{1}{2}}}\left(d_{ij}\leq d_{0}\right)$$

In the second step, take any residential point *i* as the demand point, and take the limit distance of people to medical institutions $d_{0}$ as the radius, establish the search domain *I*. Then find all the medical institutions *j* in the search domain, and the supply and demand ratios of these medical institutions $R_{j}$. Then, the supply–demand ratio of these medical institutions is summed up based on Gaussian decay function to obtain the accessibility $A_{i}^{D}$ of residential point *i* based on the distance cost. The higher the value of $A_{i}^{D}$, the higher the degree of accessibility.
(5)$$A_{i}^{D}=\underset{j\in \left\{d_{ij}\leq d_{0}\right\}}{\sum }G\left(d_{ij}\right)R_{j}$$

## 3. Results

### 3.1. Spatial Distribution Characteristics of Medical Institutions in Guangzhou

The kernel density values of medical institutions in Guangzhou are shown in Figure 2. The figure shows an apparent aggregation of medical institutions in YX, LW, HZ, and TH with high values and a contiguous aggregation. High-value agglomeration areas in the remaining districts are scattered. Taken together, this suggests that the spatial distribution density of medical institutions in southern part of Guangzhou is stronger than that in the northern part.

According to the result of the SDE analysis, the standard ellipse azimuth of primary, secondary, and tertiary hospitals’ medical institutions is all in the range of 159–178°. Although the azimuth of unclassified medical institutions is only 8.5°, it does not change the overall distribution direction of medical institutions in Guangzhou because the number of unclassified medical institutions is small. The standard elliptical azimuth of all medical institutions is also in the range of 159–178°, indicating that the spatial layout of medical institutions in Guangzhou mainly shows a northwest–southeast pattern (Table 1). The short semi-axis of the standard ellipse from primary to tertiary medical institutions tends to shorten as the level increases, from 34,518.96 m in Level I to 17,918.97 m in Tertiary. The long semi-axis mainly fluctuates in the range of 17,300–19,900 m, with the long axis of Secondary medical institutions being 19,880.21 m, which is nearly 2500 m longer than that of Primary. Compared with the Secondary, the long axis of the Tertiary is shortened to a similar length to that of the Primary. According to the variation of the short and long semi-axes, it can be found that the distribution of medical institutions in Guangzhou shows a continuous contraction in the north–south direction with the increase of grade, and is more stable in the east–west direction. The flatness of the SDE gradually increases with the gradual increase and is close to 1. The flatness indicates that the medical institutions are gradually concentrated and distributed in an approximate circle with the increase of grade. The flatness rate of the overall SDE is 0.86, which means that the overall influence of medical institutions in Guangzhou is more robust in the north–south direction than in the east–west direction.

From the distribution range and trajectory changes of each grade (Figure 3), it can be seen that the spatial distribution range of high-grade medical institutions is more contracted and concentrated than that of the primary healthcare. Moreover, the influence range of most high-grade medical institutions is included in the influence range of the primary healthcare. Therefore, the SDE of primary healthcare can be regarded as obtained by stretching the high-grade in the northwest–southeast direction. Regarding the trajectory of the center of gravity, the centers of the SDE of all levels of medical institutions are located in the administrative district of TH. The centers of gravity of the ellipse of different levels do not move significantly in the north–south direction but are shifted significantly in the east–west direction. Specifically, the center of gravity of Secondary medical institutions has slightly shifted 379.0 m in the west direction compared with Primary medical institutions. However, the displacement in the north–south direction has shifted 1142.6 m in the north direction. The center of gravity of tertiary medical institutions has shifted significantly 5497.7 m in the west direction compared with primary medical institutions. However, the displacement in the north–south direction has only shifted 18.2 m in the south direction. This indicates that the three old urban areas on the west side of Guangzhou City, LW, YX, and HZ have the advantage of excellent medical resources.

### 3.2. Equality Measurement Results

The Lorenz curve based on population and regional area allocation shows that Guangzhou’s medical institution resources are closer to the absolute equity line and more equitably allocated by population than by regional area. In terms of population allocation, the overall equity of medical institutions is good. Among the equity of medical institutions at all levels, the equity of secondary medical institutions is the best, and the equity of unclassified and primary medical institutions is the worst (Figure 4). The equity of geographic area allocation is worse than that of population allocation. Among the equity of geographic area allocation, the equity of primary medical institutions allocation is the best, and the equity of unclassified and tertiary medical institutions is worse (Figure 5).

The Gini coefficients of each class of medical institutions by population and area were calculated (Table 2). The Gini coefficients of the number of institutions by population in Guangzhou were all smaller than those by geographical area. Regarding equity of allocation by population, the Gini coefficients of both resources were less than 0.4 for secondary medical institutions (0.19) and tertiary medical institutions (0.30). However, the Gini coefficients of unclassified medical institutions (0.41) and primary medical institutions (0.42), and both resources exceeded the warning line of distribution gap (0.4). In terms of equity of allocation by geographic area, the Gini coefficient of each resource exceeds 0.50, with unclassified medical institutions (0.76) and tertiary medical institutions (0.78) already in a highly unbalanced state.

### 3.3. Accessibility Measures

Regarding the different service capabilities of different medical institutions, this study set the service threshold for unclassified and primary medical institutions to 1 km, for secondary medical institutions to 3 km, and for tertiary medical institutions to 5 km [36]. The analysis was conducted with the help of ArcGIS10.2, and the results were graded and visualized using geometric intervals.

As seen in Figure 6a, the accessibility of unclassified and Primary medical institutions in Guangzhou shows a gradually decreasing distribution from the periphery of the city to the central city. The spatial differences in the accessibility of unclassified and Primary are pronounced and generally low in the central urban area. There are three high-value spatial clusters of different shapes and sizes in the western part of HD, the southwestern part of HP, and the northern part of NS, and the size of the spatial clusters increases in turn. Accessibility is related to population density, population size, and the number of medical institutions. Although the number of unclassified and first-class medical institutions in these three high accessibility areas is low, the population demand around them is also low, so their spatial accessibility is high. However, for YX, LW, HZ, and TH, which are in the low accessibility area, although they have many medical institutions, the service capacity of unclassified and first-class medical institutions is limited by the high population density. Using 1 km as the service threshold, the unclassified and Primary facilities serve 62.8% of the city’s population and 15.0% of the city’s total service area.

The spatial distribution of accessibility of secondary medical institution resources in Guangzhou is similar to that of the unclassified and primary, showing a distribution of high accessibility in the periphery of the city than in the central city (Figure 6b). Unlike the accessibility of the unclassified and primary medical institutions, there are small areas of high accessibility values in the western part of CH, central HD, northern and central BY, and central NS. The accessibility in YX, LW, HZ, and TH, where secondary medical institutions are more densely distributed, is not as high, probably due to the dense population in these areas and the high demand and insufficient service capacity of medical institutions. Using 3 km as the service threshold for secondary care facilities, secondary care facilities serve a total of 81.2% of the population of Guangzhou, with a service area of 25.7% of the total area.

The accessibility of tertiary medical institution resources in Guangzhou shows a characteristic of spreading outward from the location of tertiary medical institutions and decreasing accessibility (Figure 6c). Within the city, the accessibility of tertiary care institutions is highest in the western part of HD. Due to the dense distribution of tertiary care institutions in the closely connected Guangzhou centrum and the extensive service threshold of tertiary care institutions, the areas with high accessibility appear in a large contiguous area. Using 5 km as the service threshold of tertiary medical institutions, tertiary medical institutions serve 82.5% of the population of Guangzhou and 30.2% of the city’s total service area.

When the accessibility of unclassified, primary, secondary, and tertiary medical institutions in Guangzhou is overlaid (Figure 7), all medical institutions serve 89.8% of the population in Guangzhou and 37.0% of the total area. It can be seen that the overall accessibility of medical institutions in Guangzhou shows a clear circle structure. The high accessibility areas are mainly clustered in the central part of Guangzhou, similar to the analysis of population distribution and kernel density of medical resources distribution. It is noteworthy that the best accessibility area is located in the western part of HD, a distant suburb, due to the presence of several tertiary care institutions and the lower population density compared to the city center. Subsequently, we calculated the percentage of the population and area covered by hospitals in each district, and the results are shown in Table 3. It can be seen that HZ, LW, TH, and YX have achieved comprehensive coverage of medical resources and have better medical help. For other districts, BY, HP, and PY have less than 10% of the population with low accessibility to medical assistance, and the area occupied varies from 20–40% of the scope of each district. However, HD, NS, ZC, and CH do not have an optimistic distribution of medical resources because 10–50% of the population does not have access to convenient and quality medical and health services.

## 4. Discussion

This study investigates the spatial distribution and accessibility of BMS in Guangzhou. First, we analyzed the spatial distribution characteristics of medical institutions in Guangzhou by using kernel density analysis and SDE method. It reflects that medical institutions in Guangzhou are mainly clustered in the central part of the city and distributed in the northwest–southeast direction. Second, we drew Lorenz curves and calculated Gini coefficients by population and geographic area, which are indicators of the equity of the distribution of medical institution resources in Guangzhou by population and geographic area. Although the results only tentatively reflect the overall characteristics of healthcare resources within Guangzhou, they provide an intuitive understanding of the imbalance in the allocation of healthcare resources within the city. The result shows that high-level medical institutions have better equality than primary healthcare, and the medical equality allocated by population is better than by area. However, as identified by the Alma-Ata Declaration (https://www.who.int/teams/social-determinants-of-health/declaration-of-alma-ata, accessed on 9 September 2022, Geneva, Switzerland), primary healthcare should be the key to the goal of health for all, so it is necessary and emergent to carry out some measures to improve the equity of primary healthcare. Finally, we calculated the accessibility of medical resources in Guangzhou city. The spatial distribution of medical resources considers various factors, including achieving as much equity as possible in allocating medical resources and guaranteeing people’s access to medical services. The spatial accessibility of medical institutions refers to the convenience of citizens’ access to nearby medical services, which is influenced by both spatial and non-spatial factors. Spatial factors mainly include the travel or time costs to the target medical institutions, and non-spatial factors include the attributes of medical institutions (type, grade, size, etc.) [2,16] and resident attributes (medical preferences, economic income, etc.) [2]. The non-spatial factors include the attribute characteristics of the healthcare facility (type, grade, size, etc.) and the resident’s attributes (healthcare preference, economic income, etc.). When calculating the time required for residents to reach the nearest healthcare facility, the nearest distance method is usually chosen [18]. However, it only considers the spatial distance between the demander and the healthcare facility, ignoring other factors that may affect the demand for healthcare services, such as the capacity of the healthcare facility to provide healthcare resources.

In contrast, the G2SFCA technique used in this paper can consider both the supply and demand sides and the distance decay problem. The results show that four of the eleven districts in Guangzhou have achieved full coverage of medical institutions, namely LW, HZ, YX, and TH. Furthermore, the remaining seven districts, BY, HP, and PY, located in the suburbs, have medium accessibility. HD, NS, ZC, and CH in the distant suburbs have the lowest accessibility.

This paper finds that the accessibility of the old town and the city center of Guangzhou is better than that of the less developed suburbs and distant suburbs. This finding is in line with assessing the spatial accessibility of healthcare facilities in other cities [7,12,29,30]. Most studies show that urban centers are better than suburban areas and that plains are better than mountainous areas in terms of accessibility to healthcare facilities, which may result from economic, geographic, and social development differences [11,41]. The generally poor accessibility of healthcare resources in the suburbs of Guangzhou suggests a lack of governmental awareness of the allocation of healthcare resources to peripheral urban areas. Urban areas are less populated, less accessible, and less valuable for development, yet suburban development is essential for urban construction and relieving pressure on urban resources.

Although Guangzhou’s medical resources are generally sound, the equitable distribution of medical resources is still an issue, especially in the distant suburbs of Guangzhou. CH and ZC are located in the outlying suburbs of Guangzhou, having low economic and social development levels, and mountains and hills dominate the natural geography. These may hinder the population inflow and the establishment of medical institutions, resulting in low medical accessibility. Guangzhou is a region where abundant medical resources vary significantly from district to district. Since the outbreak of COVID-19, the demand for medical and health care resources in each district has increased, the resource gap has expanded, and the uneven distribution of resources has emerged.

This study can guide policymakers in conducting more equitable health service planning. For example, to improve the accessibility of healthcare resources in Guangzhou, the government needs to reduce the gap between different districts by increasing public healthcare investment to build sufficient medical facilities [19], improving the treatment of healthcare practitioners in remote areas to attract professional medical technician [22,29]. Additionally, improving transportation and conditions [7,19,29] in the suburbs helps promote a more balanced distribution of healthcare resources to achieve the goal of universal access to healthcare services.

Of course, the research in this paper also has some limitations. First, the service distance thresholds for each class of medical institutions used in this paper for calculating accessibility are linear distances. In contrast, the distances people usually consider for travel are road network distances, which may reduce the reliability of the results and their explanatory power. However, based on the fact that the shortest passable distance between two points is 1.2–1.4 times the straight-line distance [38], this study’s results can still reflect the accessibility of health care services in Guangzhou and provide suggestions for policy formulation. Second, in this study, only the Lorenz curve and the Gini coefficient were used to assess the degree of inequality, and subsequent studies could use more spatial econometric indicators to support these findings quantitatively.

## 5. Conclusions

Located at the southern gate of China, Guangzhou is under tremendous pressure for epidemic prevention and control. With a large population base and a concentrated mobile population, Guangzhou bears the medical service function of Guangdong Province and even the whole country. This paper analyzes the spatial distribution characteristics and patterns of medical institutions in Guangzhou through kernel density analysis, SDE, and other spatial analysis methods. The equity of medical institutions in Guangzhou was assessed by the Lorenz curve and Gini coefficient, and the accessibility of basic medical services in Guangzhou was assessed by the G2SFCA method.

(1) The distribution of medical resources in Guangzhou is uneven, with medical resources concentrated in the central part of the city and a few in the peripheral areas of Guangzhou, such as CH, ZC, HD, and NS.

(2) In terms of population and geographical distribution, there is inequity in the distribution of medical resources in Guangzhou.

(3) The accessibility of medical and health services in Guangzhou is good, but there are differences in the accessibility of different levels of medical institutions, with primary medical institutions being worse. There are also significant differences in the accessibility of medical institutions among districts, and the accessibility of areas located in the remote suburbs is worse.

(4) To reduce the differences in accessibility of medical resources among districts in Guangzhou and improve the efficiency of medical resource allocation, the government should consider demographic and geographical factors when formulating the layout plan of medical and health facilities. In addition, the government should increase financial investment in medical facilities, especially primary healthcare. Primary healthcare can guarantee that individuals and families maintain health at an affordable price, which can significantly improve the inequity and accessibility in remote suburbs and transportation conditions in remote areas to meet residents’ medical needs.

The study results reveal the spatial accessibility characteristics of graded medical facilities in Guangzhou and propose policy recommendations to improve the graded medical system in Guangzhou and the equity of medical services for residents. In addition, our study contributes to measuring the accessibility of graded medical facilities. Therefore, these recommendations can be incorporated into the consideration of other large cities in China.

## Figures and Tables

**Figure 1 ijerph-19-14656-f001:**
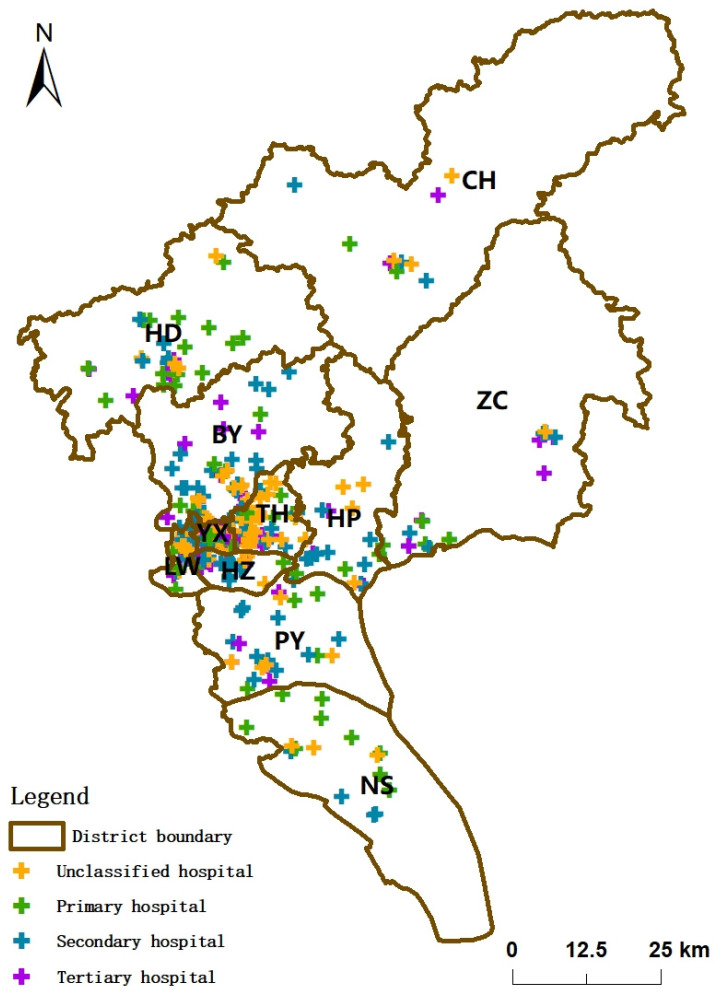
Distribution of hospitals in Guangzhou.

**Figure 2 ijerph-19-14656-f002:**
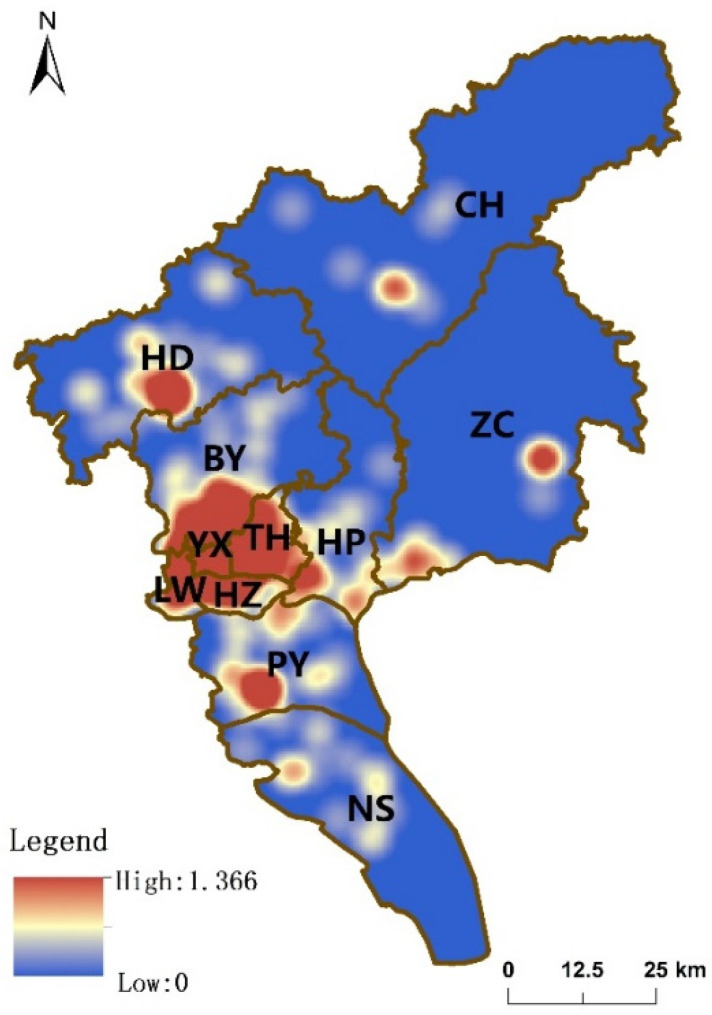
Kernel density map of medical institutions in Guangzhou.

**Figure 3 ijerph-19-14656-f003:**
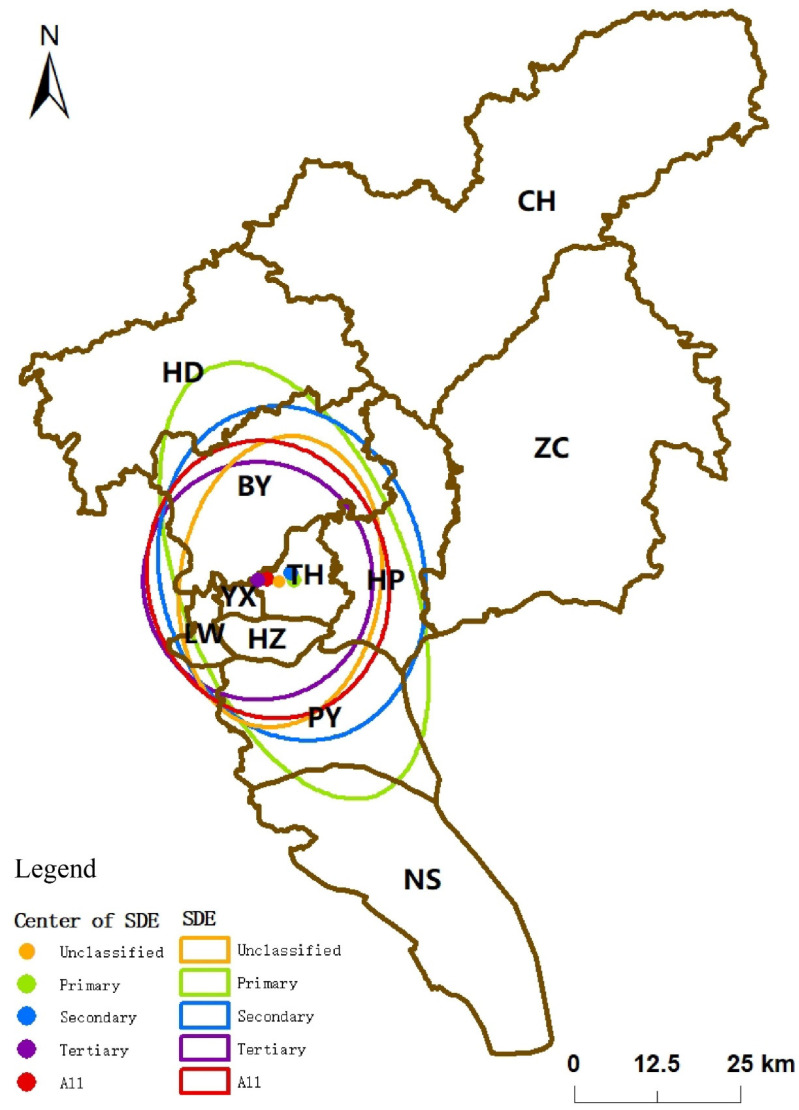
SDE spatial distribution of different levels of medical institutions.

**Figure 4 ijerph-19-14656-f004:**
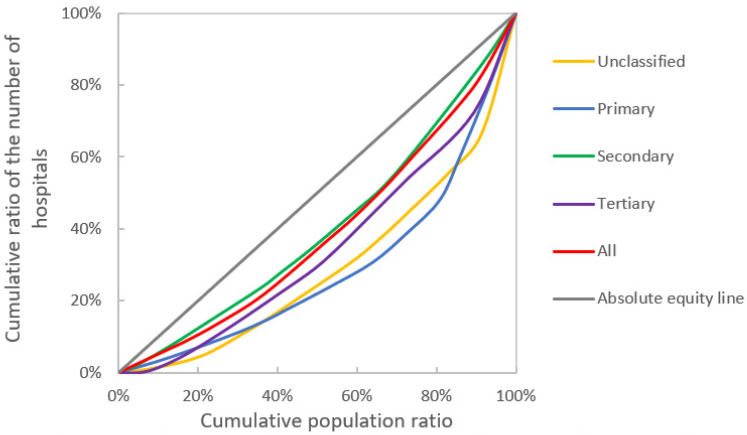
Lorenz curve for different levels of medical institutions by population.

**Figure 5 ijerph-19-14656-f005:**
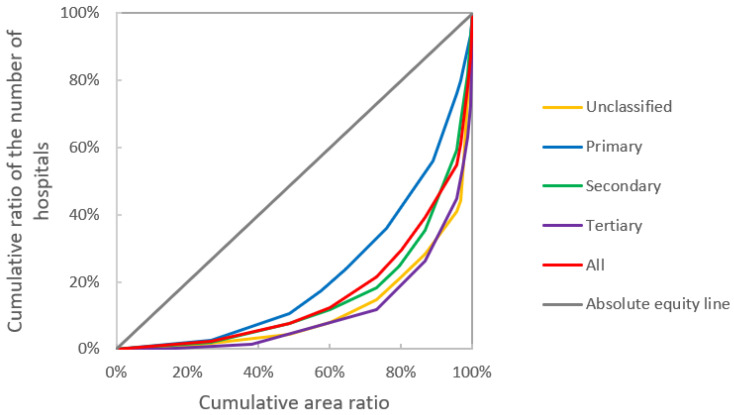
Lorenz curve for different levels of medical institutions by geographical area.

**Figure 6 ijerph-19-14656-f006:**
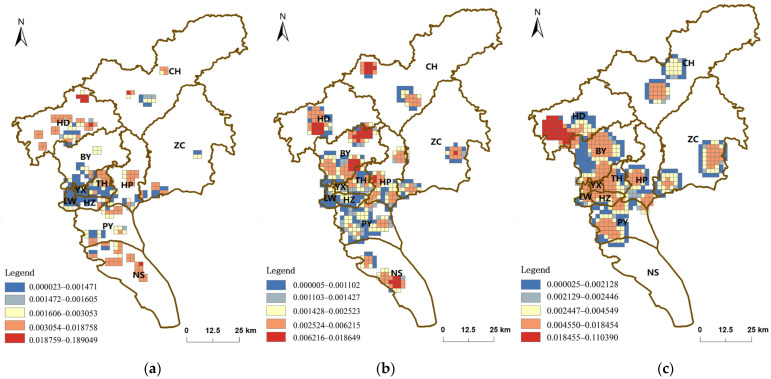
Evaluation of accessibility of tertiary medical institutions: (**a**) The accessibility of unclassified and primary hospitals; (**b**) The accessibility of secondary hospitals; (**c**) The accessibility of tertiary hospitals.

**Figure 7 ijerph-19-14656-f007:**
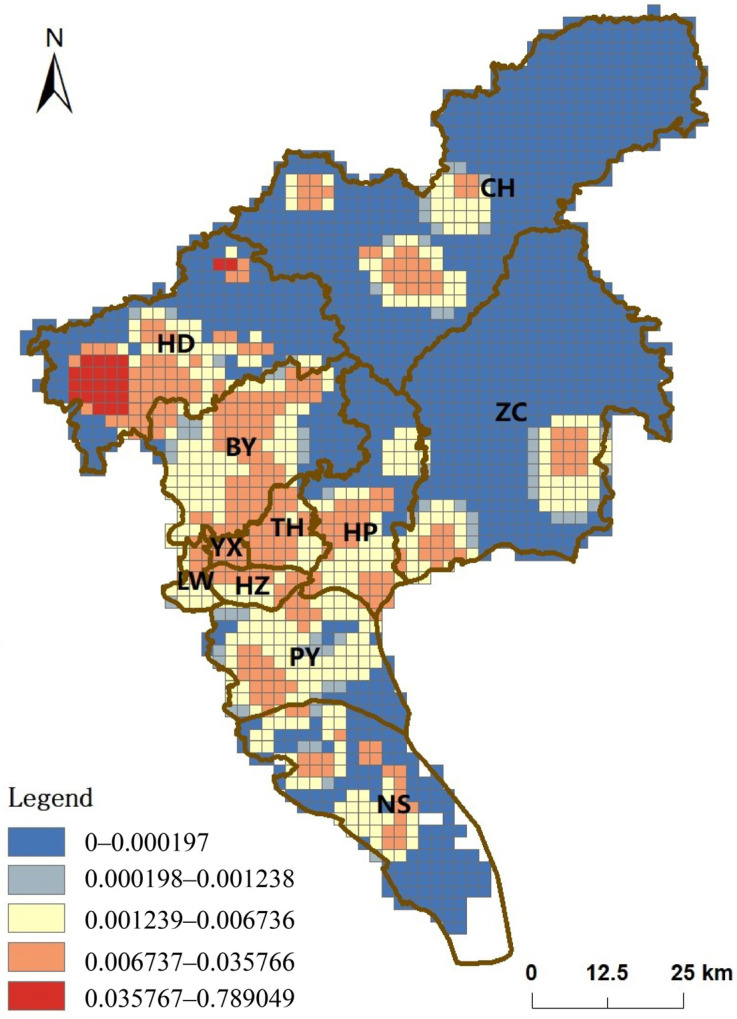
Overall medical accessibility in Guangzhou.

**Table 1 ijerph-19-14656-t001:** SDE information of different levels of medical institutions.

Medical Institution Level	Short Axis/m	Long Axis/m	Rotation Angle/°	Flat Rate
Unclassified	15,148.67	22,049.88	8.50	1.46
Primary	34,518.96	17,302.04	159.20	0.50
Secondary	25,466.40	19,880.21	166.74	0.78
Tertiary	17,918.97	17,307.33	177.64	0.97
Overall	21,070.55	18,135.32	166.70	0.86

**Table 2 ijerph-19-14656-t002:** Gini coefficient of medical institutions in Guangzhou.

Classification	Cumulative Percentage by Population	Cumulative Percentage by Area
Unclassified	0.41	0.76
Primary	0.42	0.56
Secondary	0.19	0.69
Tertiary	0.30	0.78
Overall	0.22	0.68

**Table 3 ijerph-19-14656-t003:** Percentage of population and area with zero accessibility by district.

Administrative District	Population Share	Area Share
HZ	0	0
LW	0	0
TH	0	0
YX	0	0
BY	4.23%	32.00%
HP	7.82%	31.76%
PY	8.83%	26.84%
HD	17.05%	54.85%
NS	32.98%	65.72%
ZC	33.76%	77.29%
CH	47.05%	92.32%

## Data Availability

No new data were created or analyzed in this study. Data sharing is not applicable to this article.

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
