# Peer review of "Research on the Equity and Optimal Allocation of Basic Medical Services in Guangzhou in the Context of COVID-19"

_ijerph, 2022, doi:10.3390/ijerph192214656_

Round 1

Reviewer 1 Report

The manuscript entitled “Research on the Equity and Optimal Allocation of Basic Medicine Services in the Context of COVID-19” has high potential with a clear and cohesive objective. The results bring innovative scientific contributions that enable managers’ decision-making applications in the most diverse areas (impacts on urban planning, hospital site selection, etc.).

Health equity is of particular importance to ensure well-being of all the people regardless of their social or economic status, especially under the influence of the COVID-19 pandemic. This paper addresses this problem using Guangzhou as an example, which makes the research problem more interesting as Guangzhou, an economically well-developed province, is generally considered resourceful in terms of medical services.

The research methods used included both classical methods for quantifying equity such as Gini coefficient but also include spatial methods such as Gaussian two-step floating catchment area method and Standard Deviation Ellipse Analysis, which can locate the areas within Guangzhou that lacks medical services.

Generally speaking, the research questions are significant and pertain to the current research focus. The methods used are appropriate and clarified the health inequity in big cities like Guangzhou. The conclusions are derived from the analysis, and the suggestions are useful to improve the overall health equity in Guangzhou. The expression of English sentences are generally good, but occasional mistakes do happen. The authors should go through the paper thoroughly and improve the mistakes.

I suggest minor revision. The detailed comments are as follows:

1. Line 37

Do urbanization and urban construction mean similar things? They need to be clarified.

2. Line 96-105

The description of Chinese scholars’ research is too detailed, lacks integration and summarization, simplification and necessary summary is needed.

3. Line 289

The name of figure 3 is vague, and can not explain the content of the figure. This also happened for some other figures and tables. A thorough examination is suggested.

4. Line 377-383

The discussion part lacks discussion about the result of Lorenz curves, Gini coefficients, and accessibility calculating. The relevant content needs to be supplemented.

Author Response

The authors would like to thank the Anonymous Reviewer 1 for the insightful and constructive comments. We have reviewed the comments and provided our responses herein. We truly believe that the changes suggested by Reviewer 1 will enhance the quality of the manuscript. A point-by-point response is presented below.

Point 1: Line 37

Do urbanization and urban construction mean similar things? They need to be clarified.

Response 1: Thank you for your comments. We have realized the meaning of urbanization and urban construction were repeated after your remind, so we deleted the word ‘urbanization’ to make the sentence concise. Please see Lines 47 in the revised manuscript for detailed modification.

Point 2: Line 96-105

The description of Chinese scholars’ research is too detailed, lacks integration and summarization, simplification and necessary summary is needed.

Response 2: Thank you for your comments. We think different scholars’ researches have different emphasis points, so it had better express them clearly rather than describe them in general. Please see Lines 106-118 in the revised manuscript for detailed modification.

Many scholars have improved 2SFCA to address this shortcoming according to their researches’ problems, objects and areas. Weight assignment and superposition of the service scope of hierarchical medical facilities were used to evaluate the medical accessibility of mountainous cities [36]. An optimized version of 2SO4SAI (two-step optimization for spatial accessibility improvement) was proposed to investigate the accessibility of medical care in rural areas [37]. A combination of the closest proximity method and a modified version of the 2SFCA method were used to analyze the differences in medical accessibility between minority and non-minority areas [17]. Gaussian function was introduced to model the effects of distance decay to analyze the accessibility of medical institutions above the second level [26]. In summary, existing studies have mainly measured the accessibility of BMS and analyzed the factors affecting equity, but less from the perspective of graded medical institutions to consider the equity differences of different levels of medical institutions.

Point 3: Line 289

The name of figure 3 is vague, and can not explain the content of the figure. This also happened for some other figures and tables. A thorough examination is suggested.

Response 3: Thank you for your comments. We have changed inappropriate names of some figures and tables into appropriate ones to explain the content better after a thorough examination. Please see Lines 285, 303, 314, 315 and 367 in the revised manuscript for detailed modification.

Point 4: Line 377-383

The discussion part lacks discussion about the result of Lorenz curves, Gini coefficients, and accessibility calculating. The relevant content needs to be supplemented.

Response 4:  Thank you for your comments. We have added relevent disussion . Please see Lines 394-395 in the revised manuscript for detailed modification.

The result shows high-level medical institutions have better equality compared with the low-level and the medical equality allocated by population is better than by area.

Reviewer 2 Report

The paper used quantitative and spatial statistical methods to analysis equity of the basic medical services in Guangzhou under the influence of COVID-19. The research questions addressed in the paper are meaningful and shed light on the performance of health systems under the heavy influence of the COVID-19pandemic, especially in mega cities such as Guangzhou where health services are generally considered better than other less developed cities. The results showed is interesting, as it suggested that even in Guangzhou, a modern and developed mega city, the basic medical services are unevenly distributed, and gave evidences on where the medically less covered areas are.

In general, the paper is well-written, the research questions are significant and interesting, the logic is clear, the data and methods used are appropriate, the conclusions are derived from the research performed, and the suggestions are constructive. The paper use of English language is appropriate, but still have some minor issues, as listed below:

1. In the abstract, the expression “From two dimensions of population and geographical area,” is awkward, try to change the sentence structure.

2. The spatial statistical methods used, which is an emphasis of the paper, was not apparent in the abstract, which should be made clear.

3. The expression “Compared with allocated by geographical area,” is unclear, too, and there are some redundancies in sentences.

4. In abstract, the expression “Although the per capita distribution of medical and health resources in Guangzhou is good”, is not clear. I suppose the author want to say that the quantity of per capita medical and health resources, as evidenced by basic medical services in Guangzhou is sufficient, but the spatial distribution is unequal.

5. The final sentence in the abstract, did not show the appropriate findings from the paper, and should be rewritten.

6. in Results section, the expression “At the same time, the rest of the districts have a patchy aggregation with significant zoning differences and an uneven spatial distribution of "more in the south and less in the north."” is awkward and should be rephrased using longer sentences.

7. Some of the table and figure names are not clear, such as table 1, Fig 3, and should be made understandable just from the captions, with time period, location, and research objects.

Author Response

The authors would like to thank the Anonymous Reviewer 2 for the insightful and constructive comments. We have reviewed the comments and provided our responses herein. We truly believe that the changes suggested by Reviewer 2 will enhance the quality of the manuscript. A point-by-point response is presented below.

Point 1: In the abstract, the expression “From two dimensions of population and geographical area,” is awkward, try to change the sentence structure.

Response 1: Thank you for your comments. We have changed the sentence structure. Please see Lines 13-18 in the revised manuscript for detailed modification.

The equity was analyzed using the Gini coefficient and Lorenz curve in terms of population and geographical area, respectively. Considering the medical demand and supply sides, the Gaussian two-step floating catchment area method was used to analyze the accessibility to different levels of medical institutions.

Point 2: The spatial statistical methods used, which is an emphasis of the paper, was not apparent in the abstract, which should be made clear.

Response 2: Thank you for your comments. We have added description of spatial statistical methods used. Please see Lines 18-21 in the revised manuscript for detailed modification.

The kernel density analysis and standard deviation ellipse showed that: the spatial distribution of medical and health resources in Guangzhou is unevenly distributed, and high-level hospitals and medical resources are mainly concentrated in the centrum.

Point 3: The expression “Compared with allocated by geographical area,” is unclear, too, and there are some redundancies in sentences.

Response 3: Thank you for your comments. We have rewritten the sentence to make our expression clear and easy to understand. Please see Lines 21-24 in the revised manuscript for detailed modification.

From the perspective of population, Guangzhou’s medical equity is generally reasonable. 

Point 4: In abstract, the expression “Although the per capita distribution of medical and health resources in Guangzhou is good”, is not clear. I suppose the author want to say that the quantity of per capita medical and health resources, as evidenced by basic medical services in Guangzhou is sufficient, but the spatial distribution is unequal.

Response 4: Thank you for your comments. We have rewritten the sentence to make our expression clear and easy to understand. Please see Lines 30-33 in the revised manuscript for detailed modification.

The quantity of per capita medical and health resources in Guangzhou, as evidenced by basic medical services, is sufficient, but the spatial distribution is unequal

Point 5: The final sentence in the abstract, did not show the appropriate findings from the paper, and should be rewritten.

Response 5: Thank you for your comments. We have rewritten the sentence to the appropriate findings. Please see Lines 34-37 in the revised manuscript for detailed modification.

Low-level medical institutions should be built, especially in distant suburbs, to strengthen basic medical service equity in Guangzhou.

Point 6: in Results section, the expression “At the same time, the rest of the districts have a patchy aggregation with significant zoning differences and an uneven spatial distribution of "more in the south and less in the north.” is awkward and should be rephrased using longer sentences.

Response 6: Thank you for your comments. We have rewritten the sentence to make our expression clear and easy to understand. Please see Lines 257-262 in the revised manuscript for detailed modification.

And high-value agglomeration areas in the remaining districts are scattered. Taken together, it suggests the spatial distribution density of medical institutions in southern part of Guangzhou is stronger than that in the northern part.

Point 7: Some of the table and figure names are not clear, such as table 1, Fig 3, and should be made understandable just from the captions, with time period, location, and research objects.

Response 7: Thank you for your comments. We have changed inappropriate names of some figures and tables into appropriate ones to explain the content better after a thorough examination. Please see Lines 285, 303, 314, 315 and 367 in the revised manuscript for detailed modification.

Round 2

Reviewer 1 Report

I have no comments.

Author Response

The authors would like to thank the Anonymous Reviewer 1 for the recognition.
